# Implementation of gender-based violence screening guidelines in public HIV treatment programs: A mixed methods evaluation in Uganda

Dorothy Thomas[1], Alisaati Nalumansi[2], Mira Reichman[1], Mine Metitiri[1], Florence Nambi[2], Joseph Kibuuka[2], Lylianne Nakabugo[2], Brenda Kamusiime[2], Vicent Kasiita[2], Grace K. Nalukwago[2], Timothy R. Muwonge[2], Jane Simoni[1,3], Elizabeth T. Montgomery[4,5], Andrew Mujugira[1,2], Renee Heffron[1,6]*

1 Department of Global Health, University of Washington, Seattle, Washington, United States of America, 2 Infectious Diseases Institute, Makerere University, Kampala, Uganda, 3 Department of Psychology, University of Washington, Seattle, Washington, United States of America, 4 Women's Global Health Imperative, RTI International, Berkeley, California, United States of America, 5 Department of Epidemiology and Biostatistics, University of California San Francisco, San Francisco, California, United States of America, 6 Department of Medicine, University of Alabama at Birmingham, Birmingham, Alabama, United States of America

* rheffron@uabmc.edu

## Abstract

### Background

HIV and gender-based violence (GBV) intersect to threaten population health. The Uganda Ministry of Health recommends routine GBV screening alongside HIV care but evidence detailing its implementation in HIV care settings is limited. We evaluated screening practices in public HIV clinics to generate evidence supporting GBV screening optimization.

### Methods

To evaluate GBV screening implementation in antiretroviral therapy (ART) clinics, we extracted client data from GBV registers at 12 public ART clinics in Uganda (January 2019-December 2021). We concurrently evaluated perceptions of GBV screening/referral practices by conducting in-depth qualitative interviews with providers (N = 30) and referral partners (N = 10). We contextualized quantitative findings with interview data which were analyzed using a thematic analysis approach.

### Results

During the evaluation period, >90% of providers in participating health facilities implemented GBV screening. Among 107,767 clients served in public ART clinics, providers identified 9,290 (8.6%) clients who experienced past-year physical, sexual and/or emotional GBV of whom 86% received counseling and 19% were referred to support services—most commonly to legal services. Key factors influencing GBV screening implementation included awareness of screening guidelines; client volume; and client's level of engagement in HIV

**Data Availability Statement:** Quantitative data are presented within the manuscript. Qualitative data are presented as quotations within the manuscript.

**Funding:** The Partners PrEP Program was funded by the National Institute of Mental Health of the U. S. National Institutes of Health (R01MH110296). RH was funded under award number K24MH123371 and DT under award number F31MH128080. These findings are solely the responsibility of the authors and do not necessarily represent official views of the National Institutes of Health. The funders had no role in study design, data collection and analysis, decision to publish, or preparation of the manuscript.

**Competing interests:** The authors have declared that no competing interests exist.

care. Providers and referral partners identified important benefits to clients (e.g., pursuit of justice and removal from violent environments) when referrals were successful. Key factors influencing referrals included financial constraints that limited referral partners' ability to provide services at no cost to clients and socio-cultural norms that inhibited client willingness to pursue support services.

## Conclusions

GBV screening implementation in ART clinics supports identification and referral of clients exposed to violence. The effectiveness of GBV screening may be limited by socio-cultural factors that inhibit client capacity to pursue referrals and fragmented and resource-constrained referral networks. Providers and referral partners identified allocating funds to support referrals and collaborative networking meetings as important opportunities for strengthening GBV referrals.

## Background

Gender based violence (GBV) is inextricably linked to the HIV epidemic. In Uganda, it is estimated that 39% of married women and men report having experienced physical, sexual or emotional violence in the past year [1]. Exposure to GBV has been associated with serious health consequences including depression, physical injury,increased vulnerability to HIV acquisition and poorer HIV-related health outcomes [2–5]. People living with HIV (PLWH) may have heightened risk of GBV exposure within their relationship when disclosing their HIV serostatus to their partner, taking antiretroviral medications and/or attempting to attend clinical visits for HIV management [6–9]. Evidence from Uganda estimates that one in three women and one in five men experience GBV within six months following a new HIV diagnosis [10]. GBV has also been found to undermine the HIV treatment cascade [11–15]. Specifically, experiences of GBV may threaten HIV treatment outcomes, mechanistically, via missed clinical appointments as well as poor adherence to antiretroviral therapy (ART) [11,16–18]. Thus, the identification of PLWH who have previously or are currently experiencing GBV represents a critical opportunity to mitigate the harmful influence of past and ongoing GBV upon outcomes of HIV disease progression.

Given the syndemic nature of GBV and HIV, diverse entities have recognized the importance of offering a coordinated response to GBV alongside existing HIV service offerings. The U.S. President's Emergency Plan for AIDS Relief (PEPFAR), for example, identifies strategies targeting GBV prevention as important components of the global HIV response [19,20]. In 2011, recognizing the importance of addressing issues of GBV among PLWH, the Uganda Ministry of Health—via the Consolidated Guidelines for the Prevention and Treatment of HIV and AIDS—outlined national recommendations to screen for GBV in HIV care settings [21,22]. Despite the prioritization of GBV screening both globally and in Uganda, only 10% of health providers are estimated to routinely screen for GBV [23–25]. Further, referrals to support services are believed to be hampered by low rates of GBV identification [25]. Together, these challenges represent critical barriers for comprehensively supporting individuals who may be especially vulnerable to GBV.

Enhancing the health system capacity to address challenges of GBV represents an opportunity to provide comprehensive care to and potentially reduce critical ART adherence barriers PLWH. The Consolidated Guidelines recommend screening PLWH for GBV every six months as a component of the broader HIV program and referring clients, as needed, to additional

support services [21]. In ART clinics, provider-administered screening for GBV is typically provided to clients during triage or clinical consultation. Depending on the nature of the violence that the client is experiencing, healthcare providers may offer physical examination, diagnostic testing services, clinical management of health concerns, counseling and/or referrals to non-clinical supportive services (e.g., police, emergency shelters and organizations that offer economic empowerment) [21]. Despite the anticipated benefits associated with offering GBV screening and referral to clients in ART clinics, there is very little evidence articulating 1) the extent to which these strategies are being implemented, or 2) influences on provider behaviors related to GBV screening and referral in HIV care settings. We conducted a mixed methods evaluation that aimed to deepen insights regarding the implementation of GBV screening with the intention of informing the evidence base in order to optimize providers' provision of GBV screening/referral in public ART clinics in Uganda.

## Materials and methods

### Setting

This work was conducted at ART clinics housed within 12 public health facilities with a designation of Health Centre (HC) III or greater in the Wakiso and Kampala Districts of Uganda. Notably, lower tier facilities (with a HC-II designation) provide more basic health care delivery while HC-III and greater facilities provide more specialized care including provision of ART services [26]. According to the National Health Facility Master List, 200 health facilities in Wakiso and 74 health facilities in Kampala have a designation of HC-III or greater [26].

### Overview and study design

We used a convergent parallel design, concurrently collecting qualitative and quantitative data. Quantitative data were derived from facility-based GBV registries from January 2019 to December 2021. Given our reliance on routinely collected data, we anticipated potential challenges of data availability. Thus, we leveraged a mixed methods approach to qualitatively elaborate upon and address potential gaps in quantitative data. This analysis was guided by the RE-AIM framework—an implementation science evaluation framework which identifies five dimensions that act as indicators of intervention performance including "*reach* of the program within the intended target population, *effectiveness* of the program, *adoption* of the program by target settings or staff, *implementation* of the program in terms of the consistency and quality of program delivery, and *maintenance* of program delivery and program effects [27–29]." We considered all five dimensions of the RE-AIM (Reach, Effectiveness, Adoption, Implementation and Maintenance) framework in our evaluation of the implementation of GBV screening practices within 12 ART clinics located in Kampala and Wakiso Districts (Table 1).

### Reach

We evaluated the *reach* dimension at the client and provider levels. This construct was quantitatively parameterized as the number and proportion of clients who screened positive for GBV by providers in ART clinics (i.e., the number of clients who reported GBV at screening divided by the number of clients served in ART clinics during the corresponding period). Additionally, we qualitatively assessed provider perspectives regarding GBV screening practices.

### Effectiveness

We evaluated the *effectiveness* dimension from the perspectives of the client, provider, and referral partner. Notably, *effectiveness* (at the client level) referred to the influence that

**Table 1. Components of RE-AIM evaluation framework: constructs, descriptions and data sources, as applied in this evaluation.**

| Construct and description | Questions addressed | Perspective | Data sources | Key qualitative findings, by construct |
|---|---|---|---|---|
| **Reach:** The number and proportion of clients screened positive for GBV by ART clinic providers. | How many clients in public ART clinics might benefit from GBV screening? What is the burden of GBV being identified by providers via screening in public ART clinics? What are ART clinic providers' perspectives about screening clients for GBV? (Qualitative) | Client Provider | Facility electronic medical record data outlining number of active ART clients during evaluation period. Routinely collected data in GBV screening register Interviews with ART clinic providers. | • Awareness of national recommendations to screen ART clients for GBV • High client volume • Clients level viral load status and subsequent engagement with clinical visits |
| **Effectiveness:** The number and proportion of clients offered referral services by ART clinic providers. | How many clients in ART clinics might benefit from GBV referral? How many clients in ART clinics were offered referrals to additional GBV support services? What challenges should be considered when offering supportive services to ART clients experiencing GBV? (Qualitative) | Client Provider Referral partner | Routinely collected data in GBV screening register Interviews with ART clinic providers and GBV referral partners. | •Belief that clients would receive support if connected to referral services •Limited availability of referral services •Concerns about the quality of services offered by referral partners •Socio-cultural factors that negatively influence clients' willingness to accept referrals •Clients having limited financial resources to access referral services •Referral services being too financially constrained to offer services |
| **Adoption:** The number and proportion of providers trained to offer GBV screening or referral in ART clinics. | How many providers in public ART clinics are trained to offer GBV screening? How many providers in public ART clinics are trained to offer GBV referral? | Provider | Interviews with ART clinic providers. | *This construct was not qualitatively assessed.* |
| **Implementation:** The consistency of GBV screening provision in ART clinics. | What adaptations have providers made to GBV screening provision? (Qualitative) What can be done to improve the quality of implementation for GBV screening/referral? (Qualitative) | Provider Referral partner | Interviews with ART clinic providers and GBV referral partners. | •Group health talks to concurrently "screen" many clients •Implementation of telephone GBV screening during COVID lockdown •Contact information for referral partners being made publicly available to clients in ART clinics •[Recommendation] Increase number of staff who screen and expand provider knowledge of GBV •[Recommendation] Strengthen networks between facility staff and referral partners |
| **Maintenance:** The extent to which GBV screening and referral has been consistently delivered over time. | How many ART clinics stopped offering GBV screening during evaluation period? How many ART clinics stopped offering GBV referrals during the evaluation period? Why did GBV screening/referrals stop during evaluation period. (Qualitative) | Facility Provider | Interviews with ART clinic providers. | •The COVID-19 pandemic negatively influenced screening and referral due to providers/clients having difficulty accessing the ART clinic and telephone GBV screening and social distancing measures depersonalizing screening process •Clients experienced financial constraints which made it difficult for them to access referral services •Referral partners, often supported by international funders, experienced financial constraints |

screening had on initiating referrals between GBV-exposed clients and referral partners (e.g., law enforcement, social services, economic empowerment and shelter services) who could offer greater support to clients than what was available through the health facility. Examples of the types of referees who received clients include police officers, social workers, counselors, case workers, lawyers and social welfare officers. The effectiveness construct was parameterized as the proportion of clients referred to additional services (i.e., the number of clients who received a referral to additional services among those who screened positive for GBV).

Notably, linkage systems between health facilities and referral partners are not well established. Thus, although the effectiveness dimension assesses clients who received a referral, we were unable to confirm if clients ultimately accessed and received referral services. We also qualitatively solicited provider and referral partner perspectives about barriers to successfully referring clients who require additional support services. We operationalized this construct in the aforementioned manner for two key reasons: 1) given our reliance on pragmatic data sources, we anticipated challenges linking available GBV screening data to relevant clinical outcome data, and 2) we recognize that GBV is a complex health and socio-cultural concern with linkage to referral services representing an important process through which GBV screening may influence violence persistence and/or health outcomes.

### Adoption

We evaluated the *adoption* dimension at the provider level. We parameterized this construct as the proportion of providers in ART clinics who were trained to screen or refer clients to additional services.

### Implementation

We evaluated the *implementation* dimension at the facility, provider, and referral partner levels. We qualitatively assessed adaptations that providers made when implementing GBV screening and referral in ART clinics.

### Maintenance

We evaluated the *maintenance* dimension at the facility and provider levels. We assessed the consistency with which GBV screening and referral was implemented over the course of the evaluation period. Leveraging data from interviews from referral partners and ART clinic providers, we evaluated factors that interrupted the provision of GBV screening and referral.

### Data sources, collection and analysis

We leveraged diverse data sources, using both quantitative and qualitative research methods. From August 2022 to January 2023, we collated deidentified data from facility GBV registers about ART client descriptors including gender, age, marital status, type(s) of GBV sustained (i.e., psychological, physical or sexual), services provided, and referrals offered. Register data were originally collected via client self-report during discussions with providers. During the discussions, clients were asked about violence sustained in the past 12 months with the following questions: Has anyone threatened, humiliated, shamed or otherwise caused you to feel afraid (Psychological violence)? Has anyone prevented you from coming to the health clinic, taking your medication or receiving treatment for your health (Psychological violence)? Has anyone hit, kicked, slapped, or threated you with physical harm (Physical violence)? Has anyone touched your private body parts when you did not want them to (Sexual violence)? Has anyone forced you to have sex (Sexual violence)? Has anyone refused to use a condom when you wanted them to use one (Sexual violence)? In the registry, providers indicated the services and/or referrals offered to clients. We also abstracted collated data from health facility electronic medical records with the total number of clients served in ART clinics during the period of evaluation to facilitate calculation of percentages. We summarized quantitative data using Microsoft Excel.

### Qualitative data: Interviews with providers and referral partners involved in GBV screening and referral

From August 2022 to January 2023, we also engaged 30 providers in ART clinics and 10 referral partners for investigator-developed semi-structured qualitative in-depth interviews (IDI). Using a semi-structured guide, we solicited interview participants' perspectives about influences on GBV screening and referral as well as recommendations for how to improve implementation effectiveness. Providers and referral partners were eligible to participate if they were engaged in GBV screening and/or referral service provision, 18 years of age or older, spoke English, were willing and able to provide informed consent, and were affiliated with one of the 12 participating ART clinics. We used a convenience sampling approach to identify and engage referral partners for interview participation. Specifically, we asked ART clinic providers to provide introductions to individuals or entities with whom they collaborated to effectuate client referrals for additional GBV support services. Depending on participant preference, interviews were conducted in English or Luganda. We audio recorded, translated into English (where applicable) and transcribed interviews. Transcripts were analyzed using Dedoose Software (Version 9.0.86) [30]. We used a thematic analysis approach to evaluate qualitative data [31]. The coding team consisted of three US-based female researchers, all of whom have received advanced training in qualitative research methods. We obtained written and informed consent from all participants.

### Ethics

Prior to conducting study procedures, providers and referral partners provided their written informed consent in English. This research study received ethical approval from the University of Washington Human Subjects Division (STUDY00013371), the Infectious Diseases Institute Research Ethics Committee (REF 024–2021), Uganda National Council for Science and Technology (HS1843ES) and administrators in all health facilities.

## Results

### Respondent characteristics (providers)

We conducted 30 in-person IDIs with providers in 12 participating ART clinics. Sixty-seven percent of respondents identified as female. The median amount of time providers had been providing care to clients experiencing GBV was 5 years (interquartile range [IQR] 4, 7; Table 2). Fifty-seven percent of respondents were counselors, 20% were medical officers, 13% were outreach or linkage facilitators and the remaining 10% were nurses.

### Respondent characteristics (referral partners)

We conducted 10 in-person IDIs with referral partners affiliated with the 12 participating ART clinics. Referral partner respondents were from three non-governmental organizations providing temporary shelter services and three justice/legal entities. Eighty percent of respondents identified as female, and the median amount of time that the referral partner had been supporting clients with GBV was 5 years (IQR 3, 11; Table 2). Respondents included social service providers ($n$ = 6), police or probation officers ($n$ = 3) and 1 nonprofit director.

### Reach

In 2019, 33% ($n$ = 4) of participating ART clinics had data available on clients who screened positive for GBV. In 2020 and 2021, 92% ($n$ = 11) of participating ART clinics had data

**Table 2. Demographic characteristics of interviewed providers and referral partners.**

| Characteristic | Providers (N = 30) | Referral partners (N = 10) |
|---|---|---|
| Time implementing GBV screening/referral, median (IQR) years | 5 (4, 7) | 5 (3, 11) |
| Gender—no. (%) | | |
| Male | 10 (33%) | 2 (20%) |
| Female | 20 (67%) | 8 (80%) |
| Age, median (IQR) years | 36 (30, 43) | 36 (32, 45) |
| Position—no. (%) | | |
| Medical officer | 6 (20%) | - |
| Nurse | 3 (10%) | - |
| Counselor | 17 (57%) | - |
| Outreach | 4 (13%) | - |
| Social services | - | 7 (70%) |
| Justice/Legal | - | 3 (30%) |
| Educational level attained—no. (%) | | |
| Primary | 1 (3%) | 0 (0%) |
| Secondary | 1 (3%) | 0 (0%) |
| Tertiary | 5 (17%) | 0 (0%) |
| Bachelor's degree | 17 (57%) | 9 (90%) |
| Postgraduate | 6 (20%) | 1 (10%) |

available on clients who screened positive for GBV. There were four facilities that had data available during all three of the years. From January 2019 to December 2021, a total of 107,767 clients attended clinical visits in the ART clinics reporting GBV screening data. Of these, 9,290 (9%) were identified as having experienced GBV in the past year. Facility GBV register data indicate that 86% of clients who screened positive for GBV received counseling delivered by providers in the health facility. Among clients who screened positive for GBV, 70% ($n$ = 6,492) were female (Table 3). Psychological violence was the most common form of violence disclosed in ART clinics, followed by sexual (33%) and physical (30%) GBV. Most clients (49%) who screened positive for GBV were married and 33% were never married. Ten percent of clients who screened positive for GBV were aged 17 or younger; 25% were aged 18–24; 34% were aged 25–31 and 33% were aged 32 or older. Isolating the facilities with data available from all three years, data were similar across the years, particularly in qualitative comparisons of 2019 to 2020/2021, periods prior to and during the COVID-19 pandemic.

**Provider perceptions of GBV screening implementation.** Respondents were aware of national guidelines recommending that providers implement GBV screening for all ART clinic clients. In addition to possessing awareness of screening recommendations, respondents indicated that they routinely screened most clients in ART clinics for GBV.

*The guidelines say that we should screen everyone and that is the ideal.*—Counselor, 49-year-old, Male

*Every new client who comes and every [HIV] positive patient who comes in... we screen for GBV. Because you might find out that this patient might have got HIV because of GBV.*—Clinician, 29-year-old, Female

*It is our custom that whenever you meet a client who has come for care, you screen for GBV. Whenever you interface with a client, you screen for GBV. So every time you meet a client... every client must be screened for GBV.*—Counselor, 47-year-old, Male

**Table 3. Characteristics of clients screened positive for GBV in public ART clinics in Uganda.**

| | 2019 4 facilities reporting data | | | 2020 11 facilities reporting data | | | 2021 11 facilities reporting data | | | 2019-2021 |
|---|---|---|---|---|---|---|---|---|---|---|
| | Female | Male | Overall | Female | Male | Overall | Female | Male | Overall | Overall |
| **GBV reported at screening** | 943 | 327 | 1270 | 2582 | 1133 | 3715 | 2967 | 1338 | 4305 | 9290 |
| Age (years) | | | | | | | | | | |
| <17 | 78 | 14 | 92 | 290 | 48 | 338 | 417 | 90 | 507 | 937 (10.1%) |
| 18-24 | 346 | 94 | 440 | 632 | 151 | 783 | 673 | 190 | 863 | 2086 (22.5%) |
| 25-31 | 315 | 105 | 420 | 924 | 392 | 1316 | 919 | 471 | 1390 | 3126 (33.6%) |
| 32+ | 201 | 113 | 314 | 743 | 526 | 1269 | 914 | 559 | 1473 | 3056 (32.9%) |
| **Marital status** | | | | | | | | | | |
| Never married | 324 | 115 | 439 | 794 | 311 | 1105 | 1057 | 418 | 1475 | 3019 (32.5%) |
| Married | 519 | 184 | 703 | 1270 | 621 | 1891 | 1239 | 689 | 1928 | 4522 (48.7%) |
| Separated | 49 | 9 | 58 | 377 | 151 | 528 | 417 | 131 | 548 | 1134 (12.2%) |
| Widowed | 8 | 2 | 10 | 48 | 13 | 61 | 52 | 7 | 59 | 130 (1.4%) |
| **Type of violence reported** | | | | | | | | | | |
| Sexual | 495 | 108 | 603 | 988 | 212 | 1200 | 987 | 258 | 1245 | 3048 (32.8%) |
| Physical | 299 | 115 | 414 | 747 | 441 | 1188 | 711 | 489 | 1200 | 2802 (30.2%) |
| Psychological | 748 | 229 | 977 | 1112 | 572 | 1684 | 1353 | 599 | 1952 | 4613 (49.7%) |
| **Service received** | | | | | | | | | | |
| Post-rape care | 356 | 96 | 452 | 515 | 186 | 701 | 591 | 227 | 818 | 1971 (21.2%) |
| Counseling | 896 | 303 | 1199 | 2284 | 895 | 3179 | 2491 | 1085 | 3576 | 7954 (85.6%) |
| **Referral offered** | | | | | | | | | | |
| Any | 207 | 60 | 267 | 513 | 217 | 730 | 555 | 241 | 796 | 1793 (19.3%) |
| Justice/Legal | 109 | 16 | 125 | 319 | 149 | 468 | 395 | 160 | 555 | 1148/1793 (64%) |
| Social services | 16 | 2 | 18 | 101 | 37 | 138 | 61 | 25 | 86 | 242/1793 (13.5%) |
| Medical | 76 | 39 | 115 | 83 | 31 | 114 | 98 | 56 | 154 | 383/1793 (21.4%) |
| Other | 6 | 3 | 9 | 10 | 0 | 10 | 1 | 0 | 1 | 20 /1793 (1.1%) |

Although respondents indicated that they understood that ART clients should be screened for GBV, providers cited the following as potential barriers to implementing GBV screening: if there were many clients requiring attention not all would be screened and if clients' viral load was well managed, they might be "fast-tracked" to pharmacy-only visits which would limit their opportunities to be screened for GBV during clinical visits.

*I would say [GBV screening] is supposed to be on a daily basis, but because of the overwhelming numbers [of clients], sometimes it's not done on a daily basis.*—Counselor, 29-year-old, Male

*Those clients from fast track. . . Yes. They are sometimes missed [for GBV screening] because they don't go into the counseling rooms.*—Counselor, 52-year-old, Male

Relatedly, respondents indicated that if clients' viral load was poorly managed, the client subsequently had more opportunities for clinical contact (i.e., intensive adherence counseling) during which they received more frequent GBV screening.

*And then for the clients that are not suppressing, we do it [GBV screening] four consecutive months. Because we see them for adherence counseling. And every time we talk to them. . . you keep on following up to see if it [the GBV] was really solved.*—Counselor, 29-year-old, Female

### Effectiveness

There were 107,767 clients served in ART clinics during the evaluation period. Approximately 19% ($n$ = 1,793) of the 9,290 clients who screened positive for GBV were offered referral services. The most common type of referral service to which ART clients were connected was to justice/legal (64%) entities—predominately comprised of police referrals—followed by medical services (21%) and social services (14%).

**Provider perceptions of GBV referral implementation.** Respondents shared perceptions about the benefits of connecting ART clients to referral partners who could help them address challenges of GBV beyond the context of the health system. They highlighted that in their role as providers, they were ill-equipped to address the full constellation of client concerns related to GBV and they indicated the importance of engaging the referral network to support clients beyond addressing clinical and basic psychosocial needs.

> *We cannot do everything... We must work hand in hand with other organizations that can support us in the areas that we can't handle.*—Counselor, 40-year-old, Female

> *When clients are referred for additional services, they get the services that they need which are good for them, especially services that we cannot offer them here.*—Nurse, 30-year-old, Female

> *One, when you send these people, for example, to a safe house, they don't have to go back to where they're being abused. That gives them more confidence and gives them better recovery. The shelters are places where people feel like they can recover. They're safe. They don't have to go to their primary abuser again. And this helps their mental health, their adherence to medication and they get better. Also, when you send them to a police facility, a court... They get the impression that they're going to get justice for what happened to them. So that makes them more confident.*—Medical officer, 29-year-old, Female

Providers indicated that clients sometimes faced challenges connecting to referral partners due to limited availability and low quality of services being offered outside the domain of the health facility.

> *We used to have a home which has closed. And now if we get [someone who needs safe shelter], we don't know where to send you.*——Counselor, 28-year-old, Female

> *It could be benefiting [clients] so so much, if [referral partners] gave quality services to our clients... We are not police officers. If I have referred you to police, then I expect the police to do the right thing for you, so that you are given right information, right fairness. But if I refer and the services there are not right... That's where we have a challenge... Our part we have played, but [when it comes to] other places [referral partners], we can't do much.*—Counselor, 45-year-old, Female

Respondents indicated that providers and clients collaboratively made a decision to pursue a referral. Respondents identified socio-cultural influences as potential deterrents for clients pursuing and/or being willing to accept referral services that might benefit them. Specifically, providers outlined how male authority and gendered expectations represent socio-cultural barriers to referral, e.g., women may prioritize familial, marital and/or maternal obligations over pursuit of support services to address challenges of GBV.

*I think we've got an issue of culture? There are some women being violated by a relative like a father-in-law. And you say to her, "This is a serious issue. I'm going to give you a referral letter to take to the police." But the lady says, "No, please don't do that. Let's go to the doctor and get the medicine. We shall handle this by ourselves." And you can't insist that, "No I will have to take you to the police!" She's the one to decide.*—Counselor, 52-year-old, Male

*It's just the mentality and our culture of, "You have to persevere everything in your marriage." These people tend to stay in terrible marriages where they're being abused continuously because they have children. "If I leave my children, what's going to happen to them? How do I go and stay at a safe house overnight while I have a two-year old at home?"*—Medical officer, 29-year-old, Female

**Referral partner perceptions of GBV referral implementation.** Referral partners indicated important ways in which they were able to support clients experiencing GBV. Respondents who offered shelter services described how they supported clients by removing them from environments in which they were exposed to violence. Participants also indicated that they supported individuals to emotionally reconcile with their GBV history and empowered them with knowledge and skills to avoid and more adeptly navigate future instances of violence.

*They become emotionally OK. They can cope. We give them coping skills and we also empower them with different skills. We empower them to know their rights and where to report. You know what happens with the threats that come with the violence, the sexual violence, the beatings and everything. [Survivors] are told, "You're not supposed to talk." And they don't know anything about the law. They don't know where to report.*—Counselor (Shelter services), 40-year-old, Female

Referral partners cited client- and institutional-level financial constraints as a barrier to offering clients quality support services that might address and/or help them to cope with GBV. Participants highlighted that if GBV survivors lacked financial resources to pay for testing and documentation of physical and/or sexual assaults, that might limit their pursuit of justice and potentially put them at risk of continued violence.

*The girl is saying, "[The person who raped me] is there." Which other evidence do you want, honestly? But you didn't do a swab [to test for sexual assault] which costs money and which the police don't have. For you to do a swab, you have to pay more money. And even if you do that swab, you have to pay for the DNA [analysis]. When you go to court they tell you, "You don't have substantial evidence." You see? We find ourselves in a position that we can't help them fully because trust me, the fact that this person is not arrested. . . it's going to happen again.*—Social worker (Shelter services), 36-year-old, Female

*For us. . . when someone is being beaten, you're supposed to fill a medical form. But we ask them, "Do you have money?" They presume that. . . we are conniving with the medical person [but] they don't fill that medical form for free. . . We tell them, "If you don't have the money now. . . If you're being assaulted and there's no medical form, the file cannot progress because there is no evidence." Where is the evidence of the assault? It is not there because they have not filled the medical form. So sometimes it also becomes a challenge. Maybe that's the money they're saying that maybe the police officers are asking [them for] . . . We write summons [requesting that the offender come to the police station]. Sometimes you tell them, "I only have one copy." As you're seeing we don't have a computer. What do you expect us to do? "If I have*

*one copy. . . probably you go home and photocopy and then you bring it back for me.*—Police officer, 36-year-old, Male

Similar to ART providers, referral partners also identified socio-cultural norms as important barriers to effectuating successful referrals for GBV-exposed clients. Specifically, providers noted that prevailing socio-cultural norms may prevent clients from pursuing referrals.

*The challenge we also have is men don't come and report. They don't. Because in African culture, men are presumed maybe to be the head of the family, to handle their matters. . . [Men] have that, "How will I go to the police and say that my wife beat me?" . . . Men fear coming [to the police] because of that. The culture bit of it. . . that we presume that you're a man. You die silent [ly]. You're supposed to manage it. It becomes a challenge.*—Police officer, 36-year-old, Male

*Families. . . need to report in time when the girl opens up to them. Either to the police or people who are concerned. . . The African culture of saying, "Whatever happens in a home should never be told to outsiders." [This mentality] should be thrown aside. . . when someone is suffering quietly.*—Director (Shelter services), 68-year-old, Female

## Adoption

In 2019, 63 providers were operating in the four ART clinics providing data during the evaluation period. All providers had been trained to offer GBV screening and 83% ($n = 52$) had been trained to offer GBV referral. In 2020, 198 providers were operating in the 11 ART clinics providing data during the evaluation period. Ninety-six percent of providers ($n = 189$) had been trained to offer GBV screening and 76% ($n = 151$) had been trained to offer GBV referral. In 2021, 195 providers were operating in the 11 ART clinics providing data during the evaluation period. Ninety percent of providers ($n = 175$) had been trained to offer GBV screening and 68% ($n = 133$) had been trained to offer GBV referral.

## Implementation

**Provider perceptions about adaptations made to implementation of GBV screening/ referral.** In addition to using the standardized questionnaires to screen clients for GBV, respondents indicated that group health education talks represented an opportunity to educate clients about GBV and initiate formal screening. Respondents indicated that group health education was offered daily in ART clinics and also when conducting outreach activities in the community. Providers reported that, during group health education talks, clients were encouraged to approach providers if they were experiencing GBV.

*As I am giving health education [about GBV], I tell the clients to come to me if they have experienced some form of violence at their homes.*—Outreach worker, 40-year-old, Female

*We give a health talk on GBV. We encourage those who have issues to come talk to us. This information is given every clinic day. . . at every CDDP [Community Drug Distribution Point]. When we go to the community to refill clients' medicines, we give health talks [about GBV]. When we are here at the facility, we are giving health talks [on GBV]. When we go for any community engagement, any gathering, we are giving health talks on GBV and other services. Through that, clients come out to talk about issues that are challenging them. After having provided information through the health talks, through the group sessions, they come for individual counseling sessions. In the individual counseling sessions, we have a screening tool that guides us.*—Counselor, 45-year-old, Female

Providers also described how they could post the contact information of referral partners offering GBV support services in a public location in the ART clinic so that clients could reach out for referral services directly if they chose to do so.

*We have our notes board. . . We keep putting telephone numbers for the police, child and family protection unit. Then there are other organizations that help clients. . . on that notes board, we put their numbers and as we give our health talks in the morning, we tell them, "If you're going through a problem. . . There are numbers there. You can call them to assist you in any case and [they will] come to your response."*—Clinician, 29-year-old, Female

**Provider perceptions about improving GBV screening.** Respondents indicated that ensuring adequate staffing, improving provider knowledge of diverse types of violence and simplifying standard operating procedures were potential strategies for improving GBV screening of clients in ART clinics.

*One, the staffing is a bit low. The numbers. . . Our clinic is also understaffed, but there are clinics where you can find three people. . . For you to register all those people. . . I feel like if they increased staffing, that is one thing that would improve. Two, the education sensitization. First of all, do these [providers] know what gender-based violence is?. . . For them it's unidirectional. Gender-based violence is when a man beats his wife. That is it. We need to sensitize staff on what IPV is and what GBV is. The other thing that can be done is sensitizing [providers] on how to use screening tools. . . Because there are some SOPs that are not very user friendly for people in lower health clinics [HC II]. If some things could be made more basic for everyone to be able to use, this would be better.*—Medical officer, 29-year-old, Female

Respondents also suggested additional trainings to ensure that providers were knowledgeable about the diverse types of GBV so that they could more effectively identify clients experiencing GBV.

*I think intensifying the information about [GBV] needs to be done. . . sensitization on how to identify more cases and how to handle sexual [assault cases]. . . I know there's a lot of knowledge gap. . . I think you can do more and better than this in case identification. I would need more training on how to handle other forms of violence. Because there are some things that I was told about that are considered as violence which I didn't know about. I think reading more about it and getting further trainings would be so good for me and the team.*—Counselor, 32-year-old, Male

**Provider perceptions about improving GBV referral.** Respondents identified 1) facility-based financial resources that could be provided to clients to pursue referral services if they lacked the means to do so, 2) collaborative meetings between ART clinic providers and referral partners, and 3) updated contact information for referral partners as potential strategies for improving GBV referrals.

*I think [relationships with referral partners] need to be strengthened through maybe. . . a joint seminar or training. Then I think that can also help us to bond more because you might find that I have never met the person, but we communicate on the phone. . . It can help us to bond more and improve and intensify GBV identification and referral.*—Counselor, 32-year-old, Male

*We need those proper, proper, proper referrals to be active. For them to actively receive our survivors and even give all the services that they need. In case there's a change [in staff] . . . But you have the old number. Let us create that strong network. . . That they know us, and we know them. We actually advocate that we, at one point, should have a meeting with all partners in the referral networks. We should know each other. . . and we get to have this mutual understanding because GBV cannot end just like that <snaps fingers>.*—Counselor, 40-year-old, Female

**Referral partner perceptions about improving GBV referral.** Respondents identified a need for greater financial resources and collaborations with health facilities as potential strategies for improving their ability to support GBV clients referred by ART clinics.

*Of course, funding. Yeah, we need funds. We need a lot of material in the counselling room. We also need resources to make them feel safe. You need to create a good environment for them. They face different issues and GBV is brought about by different. . . It can even be lack of food in the home that will bring GBV. People have not been eating. . . Then you bring this girl into a safe space, when you have the funds to take care of them and equip them with their skills, you are good to go.*—Counselor (Shelter services), 40-year-old, Female

*I think we have to have continuous coordination meetings, interactions, interface. We need to plan together. We need to have engagements. They need to focus so much on working with us. Because some do not. Their focus has to be working. Let's shoot in the same basket. Working in a coordinated way.*—Probation officer, 47-year-old, Female

*I think we need to know how we can all work together to help the families where this person has come from. We need to network to see how we can either apprehend the guy who has done this. . . work together to see how this girl can be resettled, work together to see how we can even do the follow-up. That we don't forget and throw this girl totally back into the fire or in freezing unsafe place. How could we follow her up and see how well she's mending?. . . I'm like, "Okay, let's work it. Let's work together. Let's see if we can do this!"*—Director (Shelter services), 68-year-old, Female

## Maintenance

Out of 12 facilities, three (25%) reported having stopped GBV screening during the evaluation period. Two (17%) facilities reported having stopped GBV referrals during the evaluation period. In all instances, the COVID-19 pandemic was cited as the cause of GBV screening/ referral interruptions in 2020.

**Provider perceptions about interruptions in GBV screening.** Respondents indicated that the COVID-19 pandemic influenced GBV screening due to government-instituted lockdowns and curfews that were put in place to slow the spread of SARS-CoV-2. Lockdown and curfew measures made it difficult for clients to attend clinic where they might be screened for GBV. Similarly, providers indicated that transportation costs (which increased substantially during the COVID-19 pandemic) were prohibitively expensive for clients and inhibited their ability to attend clinical visits where they might be screened for GBV. Additionally, providers indicated that fewer staff presented physically to work in order to support physical distancing and promote the health and safety of clients as well as fellow providers. During the pandemic, respondents indicated that GBV screening occurred in-person with limited frequency and that telephone-based GBV screening was introduced during the pandemic.

*Screening was limited. And the turn up of clients was also limited during COVID. And you would not identify as many cases as you would. . . if there wasn't COVID.*—Counselor, 32-year-old, Male

*During COVID? Actually, we are not really screening because there was a time when we were not coming here at the facility. We were at home. . . But even clients were not even coming, because we are not moving during lockdowns. And then it was even very expensive, so people actually were not coming. We were in fear of getting infected with COVID.*—Counselor, 32-year-old, Female

Providers also indicated that physical distancing complicated having intimate discussions and that telephone screening could complicate the clients' capacity to disclose experiences of GBV. These factors were also believed to influence GBV screening.

*We could not be in close contact with the clients. And distance also matters for someone to disclose to you. Those days of COVID. . . some of us [providers] are not even working at the facilities. We could work from home. You screen somebody using a phone and it is not easy. They cannot easily disclose. . . Sometimes in their surroundings they'll fear to disclose. Sometimes you can call them when the husband is there. And yet the husband is the culprit, so they cannot give you the details.*—Counselor, 42-year-old, Male

**Provider perceptions about interruptions in GBV referral.** Respondents identified the aforementioned barriers to GBV screening as factors that negatively influenced GBV referrals. Additionally, providers indicated that clients were finically burdened by transportation costs which represented an important barrier to GBV referrals during the pandemic.

*Because there was not much screening. Now if I am not screening cases what will you refer? Can you refer zero? <laughs>*—Clinician, 29-year-old, Female

*At the facility, few health workers were available, so case identification was limited which was also challenge. Those that experienced sexual violence, they couldn't seek PEP [Post-exposure prophylaxis] services because there was no transport means. Then those that needed further assistance from legal aid or health facilities for further management. . . they couldn't transport themselves because of transport means that were more [expensive] or [not operating] during the lockdowns.*—Counselor, 32-year-old, Male

Respondents reported that some referral partners were unable to accept clients during the pandemic. Providers also indicated that referral partners faced resourcing challenges that inhibited their ability to refer ART clinic clients for support.

*You may find that in maybe 2021, we were not referring people, apart from to the police. . . For the other referrals, we were not carrying out those referrals. Because of COVID most of the organizations were constrained [financially]. They are basically just recovering.*—Counselor, 32-year-old, Female

## Discussion

To our knowledge, this is the first study of its kind to evaluate the implementation of GBV screening in Ugandan public HIV treatment programs. This mixed-methods evaluation featured quantitative data from GBV registers and qualitative data from interviews with providers

as well as referral partners charged with supporting GBV-exposed clients. Qunatiative data revealed that routine GBV screening in ART clinics facilitated the identification of GBV-exposed female and male clients—although the prominence of GBV exposure was perhaps less pronounced than might be expected for a group of individuals believed to have enhanced vulnerability to violence. Qualitative data underscoring provider and referral partner perspective-silluminated context-specific barriers to GBV screening and referral while emphasizing the importance of strengthening communication channels and collaborating to strengthen referral networks and better support clients experiencing GBV.

Past-year GBV was identified in fewer than 9% of clinical ART visits which was an unexpected observation given existing evidence suggesting a relatively high prevalence of past-year GBV in Uganda [1,32]. This finding was further unexpected given that individuals living with HIV may be especially vulnerable to experiences of GBV compared to their HIV-negative counterparts [10–15]. One potential source of this discrepancy may be overreporting of clients screened for GBV. For example, provider-adapted screening methods (e.g., telephone-based or group screening) may be less effective than in-person screening at identifying GBV-exposed clients. Additionally, clients with a GBV history may be reluctant to disclose in HIV treatment settings perhaps due to lack of confidence that they would receive meaningful support to address their issues of relationship violence [33]. Thus, client reluctance to disclose experiences of violence may further contribute to underestimates of GBV among ART clinic clients.

Similar to prior research, our findings indicate that relatively few clients received referrals to additional services after having screened positive for GBV in public health settings [34,35]. Several potential factors may have contributed to the low number of referrals observed in this evaluation. Notably, the screening instrument inquires about all GBV exposures during the past year. Thus, if clients perceived their experience(s) of GBV to no longer be an active threat because it happened earlier in the year, then they may not have considered referral services as particularly relevant. Finally, in qualitative interviews, respondents revealed positive perceptions about how referring clients to additional services might help them to address their issues of violence more comprehensively; however, providers also expressed concern with the quality of referral services as well as financial and socio-cultural constraints that threatened clients' ability to benefit from referral services. Expectancy Theory posits that providers will be more motivated to refer clients if they expect that their actions will lead to positive results [36,37]. If providers had low expectations about clients receiving help from referral partner, that might have reduced their likelihood of offering referrals. Thus, improving provider perceptions that clients will get the help that they need might enhance provider willingness to initiate referral services. In qualitative interviews, both providers and referral partners underscored the importance of strengthening referral networks which may foster better interactions between providers and referral partners, better expectations about services offered to better support clients, and identification of opportunities to address implementation gaps and barriers to success.

While the presented findings offer salient and pragmatic insights regarding the implementation of GBV screening in public ART clinics, it is prudent to contextualize these findings by considering their limitations. First, we were unable to assess whether clients who were screened for GBV in the 12 participating facilities were indeed supported to reduce, prevent, or eliminate the violence that they were experiencing as client violence outcomes were not typically entered into GBV registers and client follow-up is not specifically related to GBV resolution. Relatedly, we were unable to assess if and to what extent screening ART clients for GBV ultimately influenced their HIV-related health outcomes. Second, the facilities included in this evaluation were designated as Health Centre III or greater and these findings may not be generalizable to lower tier health facilities that may have less resourcing and capacity to implement routine GBV screening. Third, although providers indicated that clients were screened for

GBV during the evaluation period and we worked closely with facility staff to identify missing registers, several facilities had missing GBV registry data—especially for 2019. We leveraged routine data sources from public health facilities (rather than research-oriented data collection methods) for this evaluation. The use of routinely collected data simultaneously represents a potential strength (e.g., we may be able to generalize findings to similar settings and data environments) as well as barrier. Our findings revealed an unstable data environment with gaps that made it challenging to present complete data. Unfortunately, the manner in which GBV screening data are recorded in the GBV registry makes it exceedingly difficult to corroborate providers' perceptions that screening is offered to all clients. The paper-based information management system that informed this evaluation necessitated a cumbersome and substantial temporal investment for abstracting data from GBV registers. When planning this study, we anticipated some challenges of data availability and, thus, elected upfront to incorporate qualitative data in order to contextualize potential gaps in quantitative findings.

## Conclusions

Our work contributes rich and contextually grounded findings to the extant evidence base describing the implementation of GBV screening implementation in Ugandan public health HIV care settings. The syndemic nature of GBV and HIV, as well as their pronounced prevalence in Uganda underscores the urgency for optimizing the health system response to GBV prevention. Further, the environment in Uganda is enabling as current guidelines support the routine provision of GBV screening in HIV treatment settings [21]. Key recommendations that emerged from this work include 1) providing funds (i.e., to facilities to support clients in accessing referral services and to referral partners to cover operational costs) to address resource constraints that represent a critical bottleneck to successful referrals, and 2) establishing opportunities for health facility staff and referral partners to interact and collaborate to reinforce the referral network. Additional research is required to assess the impact of GBV screening and referral on violence resolution and/or client HIV-related outcomes including engagement in care, ART adherence and viral suppression. Future work to assess adaptations (e.g., screening conducted over telephone, group screenings and client-initiated referrals) that providers make to GBV screening as well as their effectiveness will also be critical for identifying and optimizing implementation strategies to address GBV in HIV treatment settings.

## Supporting information

**S1 Checklist.**
(DOCX)

## Acknowledgments

This work would not have been possible without the gracious support of the twelve participating health facilities as well as their respective staff. We extend our deepest thanks to study participants who generously offered their time to share their experiences supporting people who are struggling with relationship violence.

## Author Contributions

**Conceptualization:** Dorothy Thomas, Renee Heffron.

**Data curation:** Dorothy Thomas.

**Formal analysis:** Dorothy Thomas, Alisaati Nalumansi, Mira Reichman, Mine Metitiri.

**Funding acquisition:** Dorothy Thomas, Renee Heffron.

**Investigation:** Dorothy Thomas, Alisaati Nalumansi, Mira Reichman, Florence Nambi, Joseph Kibuuka, Lylianne Nakabugo.

**Methodology:** Dorothy Thomas, Jane Simoni, Elizabeth T. Montgomery, Andrew Mujugira, Renee Heffron.

**Project administration:** Dorothy Thomas.

**Supervision:** Dorothy Thomas.

**Validation:** Dorothy Thomas, Alisaati Nalumansi, Mine Metitiri, Florence Nambi, Joseph Kibuuka, Lylianne Nakabugo, Brenda Kamusiime, Vicent Kasiita, Grace K. Nalukwago, Timothy R. Muwonge, Jane Simoni, Elizabeth T. Montgomery, Andrew Mujugira, Renee Heffron.

**Writing – original draft:** Dorothy Thomas.

**Writing – review & editing:** Dorothy Thomas, Alisaati Nalumansi, Mira Reichman, Mine Metitiri, Florence Nambi, Joseph Kibuuka, Lylianne Nakabugo, Brenda Kamusiime, Vicent Kasiita, Grace K. Nalukwago, Timothy R. Muwonge, Jane Simoni, Elizabeth T. Montgomery, Andrew Mujugira, Renee Heffron.

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
