## [Decision Letter · Decision Letter 0]

21 Feb 2024

PGPH-D-23-00915

Implementation of gender-based violence screening guidelines in public HIV treatment programs: A mixed methods evaluation in Uganda

Dear Dr. Heffron,

Thank you for submitting your manuscript to PLOS Global Public Health. After careful consideration, we feel that it has merit but does not fully meet PLOS Global Public Health’s publication criteria as it currently stands. Therefore, we invite you to submit a revised version of the manuscript that addresses the points raised during the review process.

We look forward to receiving your revised manuscript.

Kind regards,

Abhijit Nadkarni

Academic Editor

Journal Requirements:

2. Please include a complete copy of PLOS’ questionnaire on inclusivity in global research in your revised manuscript. Our policy for research in this area aims to improve transparency in the reporting of research performed outside of researchers’ own country or community. The policy applies to researchers who have travelled to a different country to conduct research, research with Indigenous populations or their lands, and research on cultural artefacts. The questionnaire can also be requested at the journal’s discretion for any other submissions, even if these conditions are not met.  Please find more information on the policy and a link to download a blank copy of the questionnaire here: https://journals.plos.org/globalpublichealth/s/best-practices-in-research-reporting. Please upload a completed version of your questionnaire as Supporting Information when you resubmit your manuscript.

3. Please provide separate figure files in .tif or .eps format only and remove any figures embedded in your manuscript file. Please also ensure all files are under our size limit of 10MB.

Additional Editor Comments (if provided):

Reviewers' comments:

Reviewer's Responses to Questions

**Comments to the Author**

1. Does this manuscript meet PLOS Global Public Health’s publication criteria? Is the manuscript technically sound, and do the data support the conclusions? The manuscript must describe methodologically and ethically rigorous research with conclusions that are appropriately drawn based on the data presented.

Reviewer #1: Yes

Reviewer #2: Yes

2. Has the statistical analysis been performed appropriately and rigorously?

Reviewer #1: No

Reviewer #2: Yes

3. Have the authors made all data underlying the findings in their manuscript fully available (please refer to the Data Availability Statement at the start of the manuscript PDF file)?

Reviewer #1: Yes

Reviewer #2: Yes

4. Is the manuscript presented in an intelligible fashion and written in standard English?

Reviewer #1: Yes

Reviewer #2: Yes

5. Review Comments to the Author

Reviewer #1: 1)The sample size, comprising 12 clinics, might restrict the ability to generalize to the district and national levels. It's important to note that there is a total of [mention the total number] ART clinics within the Wakiso and Kampala Districts of Uganda, for context. However, despite this limitation, the study provides valuable insights into the specific clinics and offers an extension of understanding within the district.

2)The clarity regarding the utilization of a questionnaire or checklist as screening tools for GBV and the methodology for segregating the prevalence of GBV among HIV clients is lacking. It is essential to include details about the specific checklist employed and the categorization of identified GBV cases. It's particularly helpful if the study refers to any recommended standardized checklist. This would enhance the comprehension of the methodology and data collection process.

3)It's not explicitly clarified the types of services, professionals, or trained staff to which clients were referred. Further elaboration is required for comprehensive understanding.

4)The study lacks a comparison between pre-COVID-19 and post-COVID-19 screening periods, which could offer valuable insights. Examining this comparison could be particularly intriguing, given reports from various studies indicating an increase in GBV cases during lockdowns, especially in terms of physical and mental violence by intimate partners. Exploring any potential influence of COVID-19 on incidence rates could provide a comprehensive understanding of the situation.

5)The study outlines factors affecting GBV screening, like awareness, client volume, and engagement. Providers and partners acknowledge benefits like justice and safety from referrals. But, the study doesn't explicitly link or weigh these factors. The impact of financial constraints and socio-cultural norms on referrals lacks clarity. Clarifying these connections would strengthen the study's significance.

Reviewer #2: The manuscript and the ways of analysing the data used were very good, the information from the research ethics committee was available, the data is great for creating public policies in the region to tackle gender violence.

6. PLOS authors have the option to publish the peer review history of their article (what does this mean?). If published, this will include your full peer review and any attached files.

**Do you want your identity to be public for this peer review?** For information about this choice, including consent withdrawal, please see our Privacy Policy.

Reviewer #1: **Yes: **Dr Md Abul Hasan

Reviewer #2: **Yes: **Pedro Renan Nascimento Barbosa

---

## [Decision Letter · Decision Letter 1]

17 Apr 2024

Implementation of gender-based violence screening guidelines in public HIV treatment programs: A mixed methods evaluation in Uganda

PGPH-D-23-00915R1

Dear Dr. Heffron,

We are pleased to inform you that your manuscript 'Implementation of gender-based violence screening guidelines in public HIV treatment programs: A mixed methods evaluation in Uganda' has been provisionally accepted for publication in PLOS Global Public Health.

Best regards,

Abhijit Nadkarni

Academic Editor

Reviewer Comments (if any, and for reference):

Reviewer's Responses to Questions

**Comments to the Author**

1. If the authors have adequately addressed your comments raised in a previous round of review and you feel that this manuscript is now acceptable for publication, you may indicate that here to bypass the “Comments to the Author” section, enter your conflict of interest statement in the “Confidential to Editor” section, and submit your "Accept" recommendation.

Reviewer #1: All comments have been addressed

2. Does this manuscript meet PLOS Global Public Health’s publication criteria? Is the manuscript technically sound, and do the data support the conclusions? The manuscript must describe methodologically and ethically rigorous research with conclusions that are appropriately drawn based on the data presented.

Reviewer #1: Yes

3. Has the statistical analysis been performed appropriately and rigorously?

Reviewer #1: Yes

4. Have the authors made all data underlying the findings in their manuscript fully available (please refer to the Data Availability Statement at the start of the manuscript PDF file)?

Reviewer #1: Yes

5. Is the manuscript presented in an intelligible fashion and written in standard English?

Reviewer #1: Yes

6. Review Comments to the Author

Reviewer #1: The author has addressed almost all of the key recommendations.

7. PLOS authors have the option to publish the peer review history of their article (what does this mean?). If published, this will include your full peer review and any attached files.

**Do you want your identity to be public for this peer review?** For information about this choice, including consent withdrawal, please see our Privacy Policy.

Reviewer #1: **Yes: **drmdabulhasan
